# Eating Disorders in the Workplace

**DOI:** 10.3390/nu17142300

**Published:** 2025-07-12

**Authors:** Nicola Magnavita, Igor Meraglia, Lucia Isolani

**Affiliations:** 1Department of Life Sciences and Public Health, Section of Occupational Health, Università Cattolica del Sacro Cuore, 00168 Rome, Italy; igor.meraglia01@icatt.it; 2Workplace Safety Prevention Service, Marche Regional Health Authority, AREA VASTA 3, 62100 Macerata, Italy; lucia.isolani@sanita.marche.it

**Keywords:** health promotion, metabolic syndrome, health literacy, stress, sleep, anxiety, depression, happiness, obesity, cardiovascular risk

## Abstract

**Background/Objectives**: Although eating disorders (EDs) affect a large portion of the population and have a significant impact on health and productivity, they are understudied in the workplace. We assessed the frequency of EDs and studied the relationship between EDs and occupational and individual factors. **Methods**: All workers undergoing health surveillance were invited to fill in the Eating Disorder Examination Questionnaire, short form (EDE-QS) and, before their routine medical examination that included metabolic tests, measure their level of health literacy, stress, quality of sleep, anxiety, depression, and happiness. Out of a total of 2085 workers, 1912 (91.7%) participated. **Results**: Suspected EDs affected 4.9% (CI95% 3.9; 5.9) of workers, with no notable difference in gender (5.3% CI95% 4.1; 6.7 in female workers vs. 4.2%, CI95% 2.9; 5.9 in male). Cases were significantly associated with trauma and emotional factors (anxiety, depression, unhappiness), but also with work-related stress and poor sleep quality, and negatively associated with health literacy. Using a hierarchical logistic regression model, suspected cases of EDs were significantly predicted in Model II by life trauma (OR 2.21 CI95% 1.40; 3.48, *p* < 0.001) and health literacy (OR 0.94 CI95% 0.90; 0.98, *p* < 0.001), in Model III also by work-related stress (OR 2.57 CI95% 1.68; 3.94, *p* < 0.001), and in Model IV by depression (OR 1.19 CI95% 1.02; 1.38, *p* < 0.05) and happiness (OR 0.88 CI95% 0.78; 0.99, *p* < 0.05). An association was also found between EDs and overweight, obesity, increased abdominal circumference, hypercholesterolemia, hypertriglyceridemia, hyperglycemia, arterial hypertension, atherogenic index of plasma, and metabolic syndrome. **Conclusions**: The workplace is an ideal setting for the prevention of EDs and their consequences. Occupational health intervention should promote health literacy, improve sleep quality, and reduce work-related stress.

## 1. Introduction

Eating disorders (EDs), better defined as eating and feeding disorders, are behavioral problems characterized by significant and persistent alterations in eating habits and accompanied by distressing thoughts and emotions [1,2]. They can also be highly complex mental illnesses associated with wide-ranging medical complications [3]. The Diagnostic and Statistical Manual of Mental Disorders, Fifth Edition (DSM-5) [4,5] classifies EDs into anorexia nervosa (AN), bulimia nervosa (BN), binge eating disorder (BED), avoidant restrictive food intake disorder (ARFID), other specified feeding and eating disorder (OSFED), unspecified feeding and eating disorder (UFED), pica, and rumination disorder. EDs are also classified by the ICD-11 [6], which adopts more extensive, but not over-inclusive, diagnostic criteria. The two schemes have convergent validity but lead to different prevalence estimates [7]. For example, an Australian study revealed that the estimated prevalence of BED in the population is higher if the ICD-11 criteria are applied [8]. EDs, which can be severe conditions that impact on physical, psychological, and social functioning, occur worldwide among females and males of all age groups, and are associated with an increased mortality risk [9]. Some systematic reviews with meta-analyses have evaluated the prevalence and incidence of different EDs [10,11,12,13,14]. These have yielded diverse estimates depending on the diagnostic criteria employed and the classification of the disorders. Studies published between 2018 and 2020 indicated a weighted lifetime incidence of 8.4% (3.3–18.6%) for women and 2.2% (0.8–6.5%) for men, and a point prevalence of 5.7% (0.9–13.5%) for women and 2.2% (0.2–7.3%) for men, with an increase from 3.5% for the 2000–2006 period to 7.8% for the period ranging from 2013 to 2018 [8]. Over time, AN rates appear to be stable, whereas there may be a decrease in BN rates [15]. Although more people are undergoing treatment, only around one-third of them have been identified by medical professionals [15]. Treating EDs can be challenging. A meta-analysis of 415 studies indicated that in 25% of cases, the disorder becomes chronic and is present after more than 5 years of follow-up. Mortality is 0.4% [16].

Such a widespread pathology implies high costs. In the United States, for the fiscal year 2018–2019, the total economic expenditure linked to EDs was estimated at $64.7 billion (95% CI: $63.5–$66.0 billion), amounting to $11,808 per affected person. OSFED constituted 35% of overall economic costs, followed by BED at 30%, BN at 18%, and AN at 17%. The decline in well-being linked to eating disorders was estimated at $326.5 billion [17]. Body dissatisfaction—a common consequence of EDs—resulted in financial and economic losses amounting to $84 billion and $221 billion due to diminished well-being. The financial implications of weight discrimination were calculated to be $200 billion, while skin-shade discrimination was assessed at $63 billion. Additionally, reduction in well-being was estimated at $206.7 billion for weight discrimination and at $8.4 billion for skin-shade discrimination [18].

Many factors may contribute to the onset of EDs, although none of them seem to play a decisive role. Adverse life experiences [19], sociocultural factors (e.g., media and peer influence, sporting or occupational activities involving body shape pressure), family factors (e.g., enmeshment and criticism), and psychological factors (negative affectivity, low self-esteem, and body dissatisfaction) were among the factors commonly reported [20,21]. Normal eating behavior is governed by a carefully regulated balance of intestinal and extra-intestinal homeostatic and hedonic mechanisms, whereas EDs are complex and maladaptive eating behaviors characterized by changes in brain–gut-microbiome interactions that result in a shift towards hedonic mechanisms [22]. Microbiota homeostasis is essential for a healthy gut–brain interaction [23]. Genetic and epigenetic factors also play a role in the development and persistence of EDs [24]. Genome-wide association studies suggest underlying metabolic dysregulation in AN [21] and polymorphism in BED [25].

EDs frequently co-occur with other mental health conditions, in particular social physical anxiety [26,27], anhedonia [28], alexithymia [29], and depression [30]. Furthermore, they are frequently associated with post-traumatic stress disorder (PTSD) [31,32,33]. An increased risk of suicide among adolescents and young adults with EDs has also been reported [34]. Besides psychiatric diseases, there are numerous medical comorbidities associated with EDs [35], including metabolic disease [36], hemicrania [37] and sleep disorders [38,39]. Individuals with BED and AN are significantly more likely to be life-time smokers than healthy controls [40]. Studies conducted during the pandemic indicate that ED symptoms have increased, probably due to both emotional eating [41], isolation, and reduced care [42]. Traumatic events and life stresses may be linked to EDs by epigenetic mechanisms [43]. An abnormal response to interpersonal stress may be one of the causal factors of EDs, but there are surprisingly few studies on this topic [44]. Health and nutrition information is also of importance since health literacy, that concerns the competencies and knowledge necessary to access, understand, and apply health indications in everyday life [45], can play a role in preventing EDs [46,47]. Social media is also thought to be a factor contributing to EDs [48,49,50].

Despite their considerable economic impact, EDs have been understudied among workers. Some research has focused on athletes or the military whose tasks require specific physical characteristics. A higher frequency of EDs has been reported in athletic activities that need certain body types [51,52,53]; however, evidence is conflicting. A systematic review with meta-analysis indicated that athletes express less body dissatisfaction in comparison to non-athletes, suggesting that sports participation may offer some protection. However, athletes who do weight-category sports (like judo) and those engaged in sports such as gymnastics and distance running that put emphasis on thinness or leanness have higher levels of disordered eating in comparison to athletes who do other sports, such as basketball and volleyball, that do not place as much emphasis on the latter aspects [54].

A frequently studied category is that of the military, which could be exposed to these disorders because of strict weight and physical fitness requirements, combat risk, and trauma [55,56,57]. Another population traditionally considered to be at high risk is that of medical students. The latter can be affected by EDs (with wide variations between studies) and have an estimated pooled prevalence ranging from 10.4% [58] to 17.4% [59] in meta-analyses. Students often struggle to cope with diagnosable EDs without being offered effective prevention or accurate diagnosis and treatment [60].

Attention to EDs in the workplace is equally poor, probably since these problems are not perceived as directly derived from work activity. However, some aspects deserve more attention: for example, the well-known association between BE [61] or AN [62] and sleep problems should encourage investigation into a possible association between EDs and night work. There is still limited evidence concerning the longitudinal relationship between eating habits and nutritional aspects, sleep dimensions (such as duration, timing, quality, and insomnia symptoms), and physical health indicators (such as anthropometric indices, fat percentage, and obesity risk) [63]. The brain regulates food intake by balancing the hedonic and metabolic pathways that determine what and how much is consumed, respectively. Additionally, the suprachiasmatic nucleus’s circadian clock centrally regulates the timings of daily feedings. When shift work disturbs this body clock, eating and metabolic issues may arise [64]. The timing and rhythm of food ingestion affect the balance between homeostatic and hedonic eating patterns. Circadian rhythm affects gut functioning, eating behavior, and microbiome interactions [65]. Studies have shown that eating at night causes a misalignment between the central and peripheral endogenous circadian rhythms that could hinder glucose tolerance [66]. Moreover, laboratory studies have demonstrated that chronic circadian disruption and sleep deprivation interfere with levels of ghrelin and leptin, peptide hormones secreted from the gut, thereby changing hunger and appetite [67]. Due to irregular eating schedules, atypical light exposure, and circadian rhythm disturbance, shift work may therefore be linked to an increased risk of obesity, diabetes, and cardiovascular illnesses [68].

Occupational stress has sometimes been associated with Eds, since these are adopted as a coping strategy [69]. A complex, gender-related association between EDs and occupational stress has been reported in obese workers [70]. Job strain increases the likelihood of BD and BMI moderates this association [71]. Occupational distress in UK doctors was reported to increase the risk of EDs, sleep disturbances, alcohol abuse, and use of drugs [72]. Moreover, there is an inverse relationship between work engagement, indicating a positive attitude towards one’s job, and disordered eating in female workers [73]. Work addiction, which constitutes a negative overcommitment to work, has been associated with the risk of EDs [74]. These reports illustrate a picture in which EDs negatively interfere with productivity. In fact, unhealthy eating habits are inversely related to the work ability of flight attendants [75]. Although the question of EDs and work clearly arouses interest, few studies have focused on this topic. A systematic review study on the impact of EDs on work performance was planned for 2022 [76] but has not yet been published.

For this reason, we decided to evaluate the prevalence of EDs in the workplace and the association of these conditions with occupational and individual factors that could play a role in the onset of disorders and possibly also in associated psychiatric, metabolic, and cardiovascular comorbidities. This study is designed to assess the impact of EDs on occupational health and to suggest lines along which prevention and health promotion programs in the workplace could be developed.

With reference to the possible relationships between variables of etiopathogenetic interest, and according to the literature, we hypothesized that:The frequency of EDs in the workplace was inversely proportional to the level of health literacy;The frequency of EDs was higher in workers who had suffered a trauma;The frequency of EDs was higher among night workers than among their daytime colleagues;The frequency of EDs was higher among smokers than among the non-smoking workers;EDs were positively correlated with occupational stress;EDs were inversely correlated with sleep quality;EDs were associated with the risk of anxiety and depression;EDs were associated with alterations in metabolic parameters and an increased atherogenic risk.

To assess the frequency of the phenomenon and its characteristics, we created a health promotion project in which all workers subjected to annual in-the-workplace surveillance were invited to participate.

## 2. Materials and Methods

### 2.1. Population

All workers exposed to occupational risks who were called for a periodic medical examination designed to prevent risks by the Catholic University of Rome, Latium (Italy) in 2022 were invited to participate in a health promotion program aimed at identifying eating disorders. Cases of suspected pathology were to be sent to the national health service for further diagnostic tests and possible treatment.

The project design was a cross-sectional census. We invited all workers to participate, without any exclusion criteria or obligation to do so. Participation was not encouraged, but it has always been very high in this type of initiative [77], since neither workers nor companies sustain any expense, because the proposed activities fall within the duties of the occupational physician. Participants signed an informed consent form and authorized the management of their data for scientific purposes as well. The results were transmitted to each employer, to the company prevention services, and to the workers’ representatives in anonymous form, together with the occupational physician’s advice on the prevention measures to be adopted in the workplace. At the end of the annual program, a total of 1912 workers out of the 2085 undergoing health surveillance responded to the invitation (participation rate 91.7%). The workers belonged to different companies from the health (1028, 53.8%), social (172, 9.0%), industrial (107, 5.6%), and commercial sectors (605, 31.6%). Female workers (1170, 61.2%) outnumbered males (742, 38.8%). Mean age was 45.63 ± 11.73.

The study was approved by the Ethics Committee of the Università Cattolica del Sacro Cuore, Policlinico A. Gemelli, Rome, on 3 March 2022 (ID 4671).

### 2.2. Questionnaire

To investigate eating disorders, we used the EDE-QS (Eating Disorder Examination Questionnaire, short form), derived from the EDE-Q (Eating Disorder Examination Questionnaire) [78], which was initially composed of 28 questions and validated in Italian [79]. For our study we used a 12-question version, which was developed to be useful in screening [80]. For each question the answer was classified according to a 4-point Likert scale ranging from 0 to 3 (for the first 10 questions, the answer ranges from zero days a week = 0 to 6–7 days a week = 3; for the last two, the answer ranges were from “not at all” = 0 to “markedly” = 3). The total score can range from 0 to 36. In this questionnaire, a score of over 15 points can indicate eating disorders [81]. In this study, Cronbach’s alpha was 0.846.

To measure Health Literacy (HL) we used the Italian version [82,83] of the Health Literacy Short Form (HLS-SF12) [84,85], a version composed of 12 re-selected questions from the 47-item European Health Literacy Questionnaire (HLS-EU Consortium, 2012) [86]. On a 4-point Likert scale ranging from “very easy” to “very difficult” (1 = very difficult, 2 = difficult, 3 = easy, and 4 = very easy), the participant was asked to respond to how easy it is to perform certain tasks, such as finding information on treatment for illnesses. The questionnaire provided a total score (ranging from 12 to 68) composed of three sub-scales (each 4–16 points) indicating therapies, prevention, and promotion. Cronbach’s alpha for HL-SF12 in this study was 0.878.

The Digital Healthy Diet Literacy (DDL) [87] was used to investigate each worker’s ability to find dietary information via the Internet. This instrument, which consists of 4 questions also graded on a 4-point scale, provides a score ranging from 4 to 16. Cronbach’s alpha was 0.918 in this sample.

Work-related stress was studied using the Italian version [88] of Siegrist’s Effort-Reward Imbalance (ERI) questionnaire [89]. For this study, we used the short version of the questionnaire [90]. The Italian version consists of 10 questions, each of which requires a 4-point Likert scale response. Three questions measure the effort that workers put into working (on a scale with scores from 3 to 12); the remaining seven questions measure the material or immaterial reward obtained from work (scores from 7 to 28). Stress is calculated as a weighted ratio between effort and reward (ERI). A score above 1 indicates distress. Cronbach’s alpha was 0.825 for effort, 0.753 for reward.

Sleep quality was assessed using the Pittsburg Sleep Quality Indicator PSQI [91]. This questionnaire consists of a combination of Likert-type and open-ended questions that allow workers to self-assess the quantity and quality of sleep. Scores for each question range from 0 to 3, with higher scores indicating sleep impairment. Seven components are generated in the questionnaire correction, respectively, related to subjective sleep quality, sleep latency, sleep duration, sleep efficiency, sleep disturbance, use of sleep medications, and daytime dysfunction. The Global PSQI Score is the sum of these component scores (range 0–21). A cutoff score of 5 for the global scale is indicated as a measure of poor sleep [92]. Reliability of the questionnaire, measured by Cronbach’s alpha, was 0.831 in this sample.

Anxiety and depression were assessed using the Italian version [93] of the Goldberg Scale (GADS) [94]. Each factor was composed of 9 binary questions, to which one point was assigned for each affirmative answer. A worker with more than five symptoms of anxiety or two symptoms of depression has a 50% chance of having a clinically important disturbance, and above these scores the probability rises sharply. The reliability of the questionnaire was 0.827 for anxiety, 0.797 for depression, and 0.893 for the whole questionnaire.

Happiness was measured according to the Abdel-Khalek questionnaire (AKQ) [95] using a single question, graded from 0 to 10. We used a one-item tool, as most studies in the literature [96] and epidemiological investigations on large populations. It has a 0.86 temporal stability, good concurrent validity, and good convergent and divergent validity [95].

During medical examinations, anthropometric data were measured according to the International Society for the Advancement of Kinanthropometry (ISAK) guidelines [97]. Participants’ height and weight were recorded in a standing position, with the head and chest aligned and arms at the sides. Measurements were taken in millimeters and kilograms. A tape measure, placed horizontally halfway between the iliac crest and the last rib, was used to measure the waist circumference of participants in a comfortable, standing position. Body Mass Index (BMI) was calculated using the formula: BMI = weight (kg)/(height (m))^2^. After participants had been in a sitting position for at least 5 min, blood pressure was measured with three successive readings and final averaging. Systolic pressure ≥ 140 mmHg, diastolic pressure ≥90 mmHg, according to the 2023 ESH European Hypertension guideline update [98,99], or continuous antihypertensive medication were considered indicators of hypertension.

Blood glucose, triglycerides, total cholesterol, and HDL cholesterol levels were determined. Cut-off levels of metabolic parameters were defined, taking into account the International Diabetes Federation (IDF) [100], the National Cholesterol Education Program Expert Panel on Detection, Evaluation, and Treatment of High Cholesterol in Adults (NCEP/ATPIII) [101], the American Association of Clinical Endocrinologists (AACE) [102], and the Joint Societies Guidelines on Management of Cholesterol [103]. Total cholesterol above 200 mg/dL (5.2 mmol/L) or HDL cholesterol below 40 mg/dL (1.03 mmol/L) in men and <50 mg/dL in women, or treatment for hyperlipidemia, were considered indices of hypercholesterolemia. Serum triglyceride level > 150 mg/dL (1.7 mmol/L) was classified as hypertriglyceridemia. A plasma glucose level > 100 mg/dL (5.6 mmol/L), or the presence of hypoglycemic drug treatment were classified as high fasting glucose. The Atherogenic Index of Plasma (AIP) was calculated as a quotient of plasma triglycerides (TG) to high-density lipoprotein-cholesterol (HDL-C) [104].

### 2.3. Statistics

We studied the distribution of the variables of interest. The Kolmogorov–Smirnov and Shapiro–Wilk tests were used to verify whether they followed the normal distribution. The sample size reassured us that parametric methods could be used even when the variables were ordinal in nature, as suggested by Lumley [105]; however, we resorted to robust methods when this assumption was not met. For this reason, we studied the correlation between the variables by calculating both Pearson’s r and Spearman’s rho. The degree of uncertainty in prevalence (Confidence Interval 95%, CI95%) was calculated by the Clopper–Pearson exact binomial test [106,107,108,109]. Before looking at how the variables related to EDs interact using a hierarchical multiple linear regression method, we made sure these variables were not too closely related to each other, as that could cause problems with the regression models. To do this, we calculated a measure called the variance inflation factor (VIF), which looks at how much the predictor variables in a regression model are related to each other [110]. In Model I, we set life trauma as the focal predictor, adjusting for age and sex. Subsequently, we introduced health literacy and digital literacy as further predictors (Model II). In Model III, we included work-related stress, and, in the final model, we also added sleep quality, anxiety, depression, and happiness (Model IV).

Statistical analyses were carried out using IBM/SPSS Statistics for Windows, Version 28.0 (IBM Corp.: Armonk, NY, USA).

## 3. Results

Ninety-three people (4.9%, CI95% 3.9; 5.9) had a score above the EDE-QS cut-off, indicating suspected EDs. None of the workers with an EDE-QS score exceeding 15 had a BMI lower than 17.5, which is consistent with a diagnosis of anorexia nervosa according to DSM-5 criteria. Most workers with suspected EDs had a BMI above 18.5, indicative of BN or BED. Two workers could have been classified as Other Specified Feeding or Eating Disorder (OSFED), based on the EDE-QS questionnaire and their BMI.

The distribution of the EDE-QS questionnaire score was non-normal (Kolmogorov–Smirnov and Shapiro–Wilk tests *p* < 0.001). The values ranged from 0 to 29, with a median value of 3. The mean value ± standard deviation was 4.73 ± 5.22. The mean score of the EDE-QS did not vary significantly across different work sectors (ANOVA = 0.420). Table 1 reports the results of the measurements carried out in the sample.

The prevalence of EDs cases did not differ across work sectors (Pearson chi square = 0.374) but was slightly higher in females than in males (62 suspected cases, 5.3% CI95% 4.1; 6.7 in female workers vs. 31 cases, 4.2%, CI95% 2.9; 5.9 in male), although the difference between genders was not significant (Pearson’s chi square = 1.234, two-sided *p* = 0.267).

Bivariate correlation analyses indicated a substantial link between most of the variables of interest (Table 2). Gender had highly significant relationships with age, eating behavior, sleep quality, and emotional factors. Compared to their male colleagues, females manifested higher EDE-QS scores (5.23 ± 5.39 vs. 3.96 ± 4.83, Student’s *t p* < 0.001, Mann–Whitney U *p* < 0.001), higher anxiety (3.11 ± 2.72 vs. 2.08 ± 2.40, *p* < 0.001), a higher risk of depression (1.98 ± 2.26 vs. 1.33 ± 1.96, *p* < 0.001), and a more elevated disordered sleep score (6.13 ± 3.48 vs. 5.49 ± 3.07, *p* < 0.001). The mean age of women was significantly lower than for male workers (44.61 ± 11.75 vs. 47.24 ± 11.54, *p* < 0.001). Aging was significantly associated with EDE-QS scores (*p* < 0.05) and inversely related to health literacy (*p* < 0.01) and digital health literacy (*p* < 0.05). Older age was also associated with higher stress (*p* < 0.01), anxiety (*p* < 0.01), and depression scores (*p* < 0.01), as well as lower happiness (*p* < 0.01), and lower sleep quality (*p* < 0.01). EDE-QS scores were positively associated with stress (*p* < 0.01), sleep problems (*p* < 0.01), anxiety (*p* < 0.01), and depression (*p* < 0.01), and inversely related to health literacy (*p* < 0.01), digital health literacy (*p* < 0.01), and happiness (*p* < 0.01). Work-related stress, anxiety, and depression were positively correlated with each other (*p* < 0.01) and inversely related to happiness (*p* < 0.01).

In our sample, 199 workers (26.1% CI95% 24.1; 28.1) were engaged in night shifts. Workers performing night shifts did not have an EDE-QS score significantly different from that of their colleagues (4.48 ± 4.90 vs. 4.83 ± 5.32, Student’s *t p* = 0.199. Mann–Whitney U *p* = 0.399).

Among the workers observed, 653 (34.2% CI95% 32.0; 36.3) were smokers. The smokers had a slightly higher EDE-QS score than non-smokers (4.77 ± 5.26 vs. 4.72 ± 5.19), but the difference was not significant (Student’s *t p* = 0.852: Mann–Whitney U *p* = 0.809). No correlation was found between the EDE-QS score and weekly alcohol consumption (Spearman’s rho = 0.02, *p* = 0.96).

A total of 349 workers (18.3% CI95% 16.5; 20.1) reported having experienced serious trauma (mourning, serious problems in the family or at work) in the previous year. These workers totaled a significantly higher EDE-QS score than the others (6.21 ± 6.12 vs. 4.41 ± 4.93, *p*< 0.001).

Using hierarchical logistic regression analysis and taking the pathological condition (EDE-QS score >15) as the dependent variable and the factors that had shown a significant correlation with EDs as determinants, we evaluated the influence of the various factors on cases of eating disorders (Table 3). All the VIF values for each of the multiple linear regression models were much lower than the usual limits that indicate a risk of collinearity (Appendix A). We can therefore exclude that there was any problem with multicollinearity in the interpretation of the results. In Model I, trauma experienced in the past year was found to be a highly significant predictor of an eating disorder, also significantly related to age. Model II indicated that health literacy very significantly reduced the risk of EDs. Occupational stress, introduced in Model III, played a significant role in generating EDs in which age was no longer relevant, but trauma maintained a significant determining effect and health literacy a protective effect. By introducing emotional and mental health factors into Model IV, the variables most closely linked to eating disorders were depression and lack of happiness, while health literacy and digital literacy maintained a positive role, even if the associations did not reach the level of significance.

BMI measurement found 12.1% (CI95% 10.5; 13.8) of workers to be obese (BMI > 30). A higher percentage (659, 34.5%) declared themselves to be overweight and 284 (14.9% of the whole sample) were undergoing treatment to lose weight. In fact, 759 workers (39.7%) had a BMI of over 25, confirming that they were overweight. Abdominal obesity was present in 316 workers (159 females and 157 males, 16.5% of the total). Workers with suspected EDs had on average a significantly higher BMI (27.89 ± 5.15 vs. 24.61 ± 4.38, *p* < 0.001), and higher waist circumference (96.27 ± 15.30 vs. 90.71 ± 12.54, *p* = 0.030) than other workers.

In the sample, 540 workers (28.2%, CI95% 26.2; 30.3) had hypercholesterolemia (total cholesterol above 200 mg/dL or HDL cholesterol < 50 in females, <40 in males, or both), or were undergoing treatment for this reason. In the whole sample, there were 148 (7.7%, CI95% 6.6; 9.0) cases of hypertriglyceridemia (>150 mg/dL). On average, workers with EDs had higher total cholesterol (207.88 ± 46.03 vs. 191.70 ± 1.62, *p* = 0.029), and higher triglyceride (103.96 ± 90.51 vs. 82.07 ± 55.89, *p* = 0.05) than their colleagues with normal dietary behavior. Among workers with EDs, there were 35 cases of hypercholesterolemia (37.6%), i.e., a significantly higher prevalence than in other workers (505 cases, 27.5%, Pearson chi square *p* = 0.039). Moreover, the prevalence of hypertriglyceridemia was significantly higher in workers with EDs than in others (18.3% vs. 7.2%, *p* < 0.001).

A total of 270 workers (14.1%, CI95% 12.6; 15.8) had hyperglycemia (>100 mL/dl) or were being treated for this problem. Hyperglycemia was significantly more frequent in workers with EDs (25 cases, 26.9%) than in other workers (245, 13.5%, Pearson chi square *p* < 0.001).

Blood pressure was altered (systolic > 140, diastolic > 90) or under treatment in 447 workers (23.4%, CI95% 21.5; 25.3). Hypertension was significantly more frequent in workers with EDs (40.9%) than in their colleagues with normal eating habits (22.5%, Pearson chi-square *p* < 0.001). Workers reporting EDs had on average higher diastolic blood pressure (82.27 ± 12.42 vs. 78.57 ± 10.93, *p* = 0.019) and non-significantly higher systolic blood pressure than their colleagues.

Finally, 225 workers (116 males and 109 females, 11.8% of the total, CI95% 10.4; 13.3) had three or more metabolic disorders and were therefore considered to have metabolic syndrome (MetS). The prevalence of MetS was significantly higher in workers with EDs (22.6 vs. 11.2, Pearson chi square *p* < 0.001). The atherogenic index AIP was significantly higher in workers with EDs than in their colleagues with normal eating behavior (2.08 ± 2.43 vs. 1.43 ± 1.27, Student’s *p* = 0.028).

## 4. Discussion

This study, which is one of the few to have investigated the potential determinants and comorbidities of EDs in the workplace, confirmed the relationship of these disorders with trauma and emotional factors, but also underlined the etiopathogenetic importance of occupational and social factors such as stress, sleep, and health literacy, thus raising the question of possible relevant health promotion intervention in the workplace. Comorbidities, consisting of mental and metabolic disorders, demonstrate the impact of EDs on health, the quality of life and productivity.

In our census, the suspected cases were almost all associated with binge eating and obesity. This was not surprising in a population whose average age is over 45 years, since many report the onset of overnutrition disorders in the peri- or post-menopausal period in women or at a corresponding age in men [111]. Studies on hospital nurses showed significant correlations between EDs, body mass index, and job performance [112]. In a large sample of over 110,000 workers, obesity and BED were associated with the highest levels of absenteeism, presenteeism, and productivity impairment [113]. A study conducted on over 22,000 US adults in the National Health and Wellness Survey indicated that BED respondents reported significantly higher rates of absenteeism, presenteeism, work productivity loss, and activity impairment than non-BED respondents [114]. Among students at a Midwestern university in the United States, 7.8% suffered from BED that was frequently associated with obesity, sleep problems, reduced classroom productivity, and daily activity impairment [115]. The impact of EDs on productivity should stimulate companies to activate prevention and health promotion.

The hypotheses we made at the beginning of the study were almost all confirmed: first, health literacy was found to be inversely proportional to the risk of EDs. This suggests that an increase in knowledge could be useful in preventing this type of disorder and in improving health in the workplace. In our case study, age was inversely correlated with literacy. Younger workers have greater access to health literacy and are better informed than older workers. This may also explain why in the sample of adults we examined, the prevalence of cases tended to increase with age. Although it is known that the initial onset of EDs occurs principally during adolescence and young adulthood [116], there is a high prevalence of recurring EDs later in life [117]. In clinical studies, the relationship between literacy and EDs has been studied especially in relation to mental health literacy [118] and social media literacy [46]. Few studies have attempted to evaluate the subjective level of skills and knowledge used by workers to understand and seek health-related information, and how this variable is related to EDs. Medical students with low health literacy tend to have lower mental health and more unhealthy eating habits than their colleagues [119]. In a large sample of the German population, poor health literacy and adverse body image were correlated with an elevated incidence of suspected EDs when compared with adequate health literacy and a more favorable body image [120]. Health literacy therefore seems to be a good way of promoting virtuous eating habits in the workplace. The more frequent onset of EDs in youth and the recurrence in adulthood suggest that prevention efforts in the workplace should be particularly aimed at younger workers, while older workers should be screened to identify and treat persistent or recurrent forms. Workplace health promotion programs should focus on increasing health literacy, especially among younger workers who are also particularly receptive to digital information. However, health checks for older workers should also be included on account of the risk of EDs in older age.

In our investigation, suspected ED prevalence was not found to be significantly associated with gender. In fact, women reported a higher mean EDE-QS score than men but did not have a significantly higher percentage of suspected ED cases. Most likely, women have a greater attention to body image and nutrition, and this justifies the higher EDE-QS score compared to men, but the cases of clinical relevance were not significantly different in the two sexes. This finding agrees with a study conducted in a German population indicating that EDs are slightly more frequent in females than in males during adolescence, but no difference is observed in adults [120]. Nevertheless, this homogeneous distribution among sexes does not exclude the potential existence of problems related to gender minorities [121,122] and gender dysphoria [123] in the workplace. Companies should carefully monitor these aspects and promote policies that prevent discriminatory behavior towards minorities.

Researchers have hypothesized that night work could be an occupational factor responsible for EDs. Our findings did not confirm this association. In our observations, however, there was an association between EDs and poor sleep quality. This apparent contradiction has some explanations. First, it is necessary to remember the difference that exists between “night worker” and “worker who performs part of the work during the night hours.” According to Italian law and scientific evidence, a night worker is someone who performs at least three hours of their daily working time between midnight and five in the morning for a minimum of eighty working days per year. In the sample we examined, there were no workers included in this definition, because the number of night shifts was lower. The alteration of biorhythms and the interactions with chronotypes that are reported in night workers are unlikely to occur in people who perform a limited number of night shifts. Furthermore, the company doctor’s decisions in previous years had excluded workers with health problems from night work. Health surveillance may have determined a healthy worker effect, since workers who work night shifts were in better health than others. This set of factors could explain why EDs were not associated with night work but with the quality of sleep. Since night workers may be induced to eat after dinner and when awake at night [124], they may develop the Night Eating Syndrome (NES), a condition described as early as 1955 and linked to obesity [125]. Although this is currently classified as OSFED, some aspects suggest that it should be treated as a separate disorder [126]. Current studies have failed to confirm this association, although it has been shown that eating meals at night increases the risk of anxiety and depression. This does not occur in workers who continue to eat daytime meals [127]. Avoiding nighttime meals prevents circadian rhythm misalignment and obesity [68], as well as hyperglycemia [66] and cardiovascular disorders [128] in night workers. Consequently, health information for night workers should consider the need to avoid nighttime meals. In our study, we failed to find an increased risk of EDs in those who performed part of their service at night. However, poor sleep quality as measured by the PSQI was found to be positively correlated with the risk of EDs and with unfavorable emotional conditions, such as anxiety, depression, and unhappiness. This observation is in line with what has been published so far. In university students there is a high prevalence of sleep disturbances, derived from eating habits, incorrect mealtimes [129], and social jetlag [130], all of which are associated with poor academic performance [131,132]. The poor sleep quality of university students has been associated with NES [133] and other EDs [134]. Poor sleep quality and disordered eating independently predicted depression [135]. The relationship between sleep and EDs is probably cyclical. Unsafe dietary intake was associated with poor sleep quality in female Japanese workers [136]. A pressing need to recover after work in addition to sleep problems were associated with poor dietary habits in Finnish workers [137].

Psychosocial stress has been frequently associated with EDs. Work-related stress in nurses was associated with EDs [138]. Job strain was associated with binge eating in the ELSA-Brazil study [139] and in an in-patient cohort of obese Caucasian workers [70]. People with high susceptibility to work stress were at higher risk of BED [140]. Excessive demands from work, social overload, and coping with negative emotions were predictors of food addiction in the German LIFE-Adult-Study [141]. Compulsive overworking is associated with EDs in Polish students [142]. In the workers we examined, occupational stress was a significant predictor of EDs. In Model III of the multivariate analysis, a much closer relationship was found between EDs and stress than with severe familiar trauma occurring in the previous year. This finding suggests that the workplace can be the ideal setting for preventing EDs by acting on occupational stressors that have a greater impact on eating behavior than life traumas to which everyone is exposed. In fact, programs for the prevention of psychosocial stress at work have often been aimed at counteracting maladaptive coping such as unhealthy eating and other unhealthy behaviors [143]. The well-known association between occupational stress and common mental disorders, including EDs [144], may also be responsible for an increased risk of occupational injury associated with disordered eating and obesity [145].

One of our “a priori” hypotheses was a possible association between EDs and smoking or alcohol consumption, but our worker census failed to confirm it. On the contrary, we found a strong correlation between EDs and anxiety, depression, and scant happiness. These associations can be interpreted in two ways. Firstly, they may be the expression of a psychopathological condition known as emotional eating resulting from depression, anxiety, and positive emotions [146,147]. Deficits in basic emotional ability are key predictors of the onset and persistence of EDs [148,149]. Indeed, many psychiatric conditions, including EDs, drug abuse, anxiety, and depression benefit from emotional regulation treatment [150]. On the other hand, EDs and addictive eating can be exacerbated by obesity and changes in body image caused by binge eating; this then creates a vicious circle in which mental health problems are both the cause and consequence of EDs [151]. In fact, mental disorders are the most well-known comorbidity of EDs. People with weight problems are also often stigmatized [152,153]. Stigmatizing workers with EDs may result in depression. A comparison of reports conducted in the UK between 1998 and 2008 have revealed that the stigmatization of individuals with EDs has diminished but still exceeds that of other mental illnesses [154].

The results collected in our occupational cohort confirm what has been found in population and military studies. A Japanese study using data from the Japanese Household Panel Survey and the Keio Household Panel Survey reported an association between low drug use, good dietary habit, and high levels of happiness [155]. EDs are a frequent comorbidity in US sailors and marines with PTSD [156]. A survey on servicewomen under 45 years in the UK Armed Forces confirmed the association between EDs, sleep problems, anxiety, and depression [157]. In Iranian military staff, unhealthy eating was associated with anxiety, depression, and stress [158]. The complexity of the relationships observed explains the difficulty encountered in treating EDs. Studies on dietitians demonstrate that many of them are reluctant to treat EDs owing to insufficient ED-specific training at university, absence of clinical guidelines, and the mental health complexities of patients affected by EDs [159]. In our opinion, the difficulty in treating EDs should encourage efforts to promote healthy eating. The workplace could represent an ideal setting for these programs due to the availability of company health surveillance services, as well as the chance to rapidly contact a large number of workers and the opportunity in larger companies to control the administration of food in company canteens [160].

In the hierarchical multivariate analysis, the introduction of depression and happiness in model IV exceeded the significance of trauma and stress in model III. This statistic could mean that these variables mediate the effect of life trauma or occupational stress. Further studies could clarify this aspect. We also found confirmation of the hypothesis that EDs were associated with metabolic alterations and alterations in cardiovascular risk indices, although it is not possible to argue whether this is an effect of diet or a result of exposure to shared risk factors. In the workers we monitored, suspect cases were mainly classified into binge-spectrum disorders (BN or BED) and had mean BMI and abdominal circumference values higher than the rest of the group. The overall prevalence of overweight and obesity in our cohort was consistent with that found among adults in the population of developed countries [161]. The association between binge-spectrum disorders and obesity is not limited to cases of addictive eating [162] or grazing (repetitive eating of small amounts of food) [163] but is very frequent in subclinical form among overweight people, and often associated with symptoms of depression and the perception of excessive stress [164]. In young adults affected by overweight or obesity, EDs are often associated with anxiety or depression [165].

Hypercholesterolemia, hypertriglyceridemia, and atherogenic risk were associated with the risk of EDs in our research, as in previous studies in the literature. Dyslipidemia was reported both in restrictive eating disorders [166,167] and in patients with BED, where it was associated with obesity and other mental disorders such as anxiety or depression [168]. The Spanish PREDIMED study showed that even in patients cured of EDs, the previous illness is associated with higher levels of BMI, severe obesity, depression, and cognitive deficit in later life [169].

In our case studies, an association was observed between EDs and hyperglycemia. This association has also been found in the literature. Eating problems have been reported in patients with type I diabetes [170,171]. US adults with EDs participating in the National Epidemiological Survey on Alcohol and Related Conditions III (NESARC-III) manifested a number of comorbidities. The main ones were depression and alcohol abuse, but hyperglycemia, hypertension, and dyslipidemia were also present [172]. Longitudinal studies have shown that BN and BED induce hypertension [173]. Our findings corroborated this association. All these associations between EDs and components of the metabolic syndrome (MetS), which, by definition, include at least three out of five disorders (obesity, hypercholesterolemia, hypertriglyceridemia, hyperglycemia, hypertension), explain the association between EDs and MetS we observed and that is present in the literature. Increased inflammation and adipokine dysfunction have been observed in adolescents with BED and obesity [174]. The Brazilian Longitudinal Study of Adult Health (ELSA-Brasil) conducted on more than 15,000 people showed that BED is followed by MetS [175]. A longitudinal study observed that after 5 years of follow-up, people with BED have a more than doubled risk of developing new components of MetS [176]. The studies showed that BED is closely linked to MetS and its components, like high blood pressure, obesity, type 2 diabetes, and unhealthy cholesterol levels, through shared behaviors, genes, and biological factors [177].

Our findings demonstrate that EDs can significantly affect the workforce and that occupational prevention and health promotion interventions can play a role in increasing health literacy, countering occupational causes, and identifying comorbidities. The considerable number of workers recruited and the high participation rate represent one of the strengths of our study. Our survey is also one of the few conducted in the workplace and probably the only one to simultaneously investigate the numerous factors associated with EDs. Nevertheless, this study has several weaknesses, the most important of which is its cross-sectional nature. The method by which we contacted all workers examined in a calendar year had the advantage of rapidly obtaining a census on different work sectors, but it prevented us from recognizing the direction of the associations observed. To interpret the data, we had to rely on the literature. Another limitation is the fact that the sample does not include all occupational categories active in the country. However, since the prevalence of EDs did not vary significantly in the different occupational sectors, this can lead us to suppose that the problem studied has similar characteristics elsewhere. An obvious limitation concerns the breadth of the EDs category. Those reported were suspected cases resulting from epidemiological investigation. The workers were warned and advised to proceed with specialist diagnostic tests; the occupational physician generally knows the outcome of these tests only one year later, on the new visit. This method involves a lack of formal diagnosis of a particular ED. An intriguing advancement in this type of study could involve conducting investigations focused on specific types of ED rather than assessing general ED risk. Finally, a limitation of the study could concern the EDE-QS questionnaire, whose psychometric characteristics have not yet been sufficiently studied. Our initial findings from factor analysis suggest that the one-factor solution often used in research is valid, but they also indicate that a multi-factor solution might provide more useful insights, especially because EDs are varied, and different factors may be more important for different disorders. We believe that this type of study could be an intriguing direction for the development on this topic.

## 5. Conclusions

EDs are relatively common among workers, since they affect 1 in every 20. These disorders are associated with trauma, emotional factors, and lack of health literacy, but also with work-related factors such as excessive stress and sleep problems. The comorbidity of psychiatric and metabolic pathologies (hypercholesterolemia, hypertriglyceridemia, arterial hypertension, hyperglycemia, obesity, metabolic syndrome, and increased atherogenic index) shows the considerable impact these disorders can have on health and productivity. The workplace is an ideal setting for health promotion actions that aim to improve health literacy and reduce occupational stress or sleep deprivation.

## Figures and Tables

**Table 1 nutrients-17-02300-t001:** Characteristics of the variables observed.

Variable [Range] (*n*)	Median	Mean ± S.D.
Age	46	45.63 ± 11.73
Eating disorders EDE-QS [0–36] ^1^ (1912)	3	4.73 ± 5.22
Health Literacy HLS [12–68] (1899)	36	37.37 ± 5.60
Digital Literacy DDL [4–16] (1889)	12	11.27 ± 2.92
Effort [3–12] (1890)	7	7.08 ± 2.34
Reward [7–28] (1890)	20	19.83 ± 3.76
Effort/Reward Imbalance [0.25–4.0] (1890)	0.83	0.89 ± 0.42
Sleep quality problems [0–21] (1898)	5	5.88 ± 3.34
Anxiety [0–9] (1903)	2	2.71 ± 2.65
Depression [0–9] (1903)	1	1.73 ± 2.17
Happiness [0–10] (1911)	7	7.12 ± 1.79

Note. ^1^ Kolmogorov–Smirnov and Shapiro–Wilk tests, *p* < 0.001. *n* = number of cases. S.D. standard deviation.

**Table 2 nutrients-17-02300-t002:** Bivariate correlation between the variables used. Spearman’s rho (upper triangle) and Pearson’s r (lower triangle).

	Sex	Age	EDs	HLS	DDL	ERI	Anxiety	Depression	Happiness	Sleep Qual.
**Sex (female)**	1	−0.104 **	0.134 **	−0.034	−0.025	−0.027	0.198 **	0.155 **	−0.044	0.085 **
**Age**	−0.109 **	1	0.053*	−0.075 **	−0.050 *	0.196 **	0.073 **	0.114 **	−0.115 **	0.138 **
**EDs**	0.119 **	0.059 **	1	−0.216 **	−0.107 **	0.202 **	0.353 **	0.340 **	−0.252 **	0.307 **
**HLS**	−0.024	−0.064 **	−0.177 **	1	0.560 **	−0.157 **	−0.232 **	−0.244 **	0.220 **	−0.201 **
**DDL**	−0.019	−0.054 *	−0.072 **	0.560 **	1	−0.065 **	−0.138 **	−0.149 **	0.140 **	−0.109 **
**ERI**	−0.038	0.167 **	0.205 **	−0.157 **	−0.066 **	1	0.422 **	0.380 **	−0.358 **	0.352 **
**Anxiety**	0.189 **	0.083 **	0.373 **	−0.232 **	−0.119 **	0.448 **	1	0.768 **	−0.488 **	0.654 **
**Depression**	0.146 **	0.116 **	0.379 **	−0.244 **	−0.144 **	0.406 **	0.783 **	1	−0.484 **	0.622 **
**Happiness**	−0.044	−0.114 **	−0.281 **	0.220 **	0.117 **	−0.369 **	−0.509 **	−0.556 **	1	−0.421 **
**Sleep Qual.**	0.093 **	0.141 **	0.345 **	−0.191 **	−0.102 **	0.366 **	0.694 **	0.653 **	−0.456 **	1

Note: The number of cases in each of the bivariate analyses ranged from 1876 to 1912. ** Correlation is significant at the 0.01 level (2-tailed). * Correlation is significant at the 0.05 level (2-tailed).

**Table 3 nutrients-17-02300-t003:** Hierarchical logistic regression. Factors associated with suspect EDs.

	Model I		Model II		Model III		Model IV	
Variable	OR (CI95%)	*p*	OR (CI95%)	*p*	OR (CI95%)	*p*	OR (CI95%)	*p*
Gender (female)	1.30 (0.83; 2.04)	0.247	1.30 (0.83; 2.05)	0.247	1.38 (0.87: 2.18)	0.172	1.03 (0.63; 1.66)	0.991
Age	1.02 (1.01; 1.04)	0.013	1.02 (1.01; 1.04)	0.016	1.02 (1.00; 1.04)	0.062	1.01 (0.99; 1.03)	0.214
Trauma	2.20 (1.40; 3.46)	0.001	2.21 (1.40; 3.48)	0.001	1.94 (1.22: 3.10)	0.005	1.21 (0.73; 2.01)	0.458
Health Literacy (HLS)			0.94 (0.90; 0.98)	0.005	0.95 (0.91; 0.99)	0.017	0.98 (0.93; 1.02)	0.326
Digital Literacy/DDL)			1.08 (0.98;1.19)	0.103	1.09 (0.99; 1.19)	0.088	1.09 (0.99; 1.21)	0.075
Stress (ERI)					2.57 (1.68; 3.94)	0.001	1.23 (0.75; 2.00)	0.417
Sleep Quality (PSQI)							1.06 (0.98; 1.15)	0.171
Anxiety (GADS)							1.04 (0.90; 1.20)	0.584
Depression (GADS)							1.19 (1.02; 1.38)	0.024
Happiness							0.88 (0.78; 0.99)	0.043

## Data Availability

Data are deposited on Zenodo https://doi.org/10.5281/zenodo.15685523. Uploaded on 17 June 2025.

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
