# Peer review of "Eating Disorders in the Workplace"

_nutrients, 2025, doi:10.3390/nu17142300_

Round 1
Reviewer 1 Report
Comments and Suggestions for Authors The paper assessed the frequency of EDs and studied the relationship between EDs and occupational and individual factors. It is a solid paper both in terms of literature overview which is up to date and refers to most important sources, as well as in terms of methodology. Moreover, it refers to important and understudied area of research related to eating disorders in the workplace, and the potential role of workplace interventions to reduce the scale and negative consequences of these problems. However, few issues should be addressed to strengthen the paper: 1) Authors refer to the DSM framework for diagnosing EDs, however, in most countries worldwide WHO's ICD is the used framework. It would be beneficial to indicate how these frameworks are similar or differ. 2) EDE-Q (Eating Disorder Examination Questionnaire) was used to measure EDs risk. However, EDs are not a uniform category and different components have slightly different weight and meaning in relation to different EDs, for example weight restraint in AN and BED. It would be recommended to show the factorial structure of EDE-Q (as to my knowledge it has not been done for its short version) to confirm that a general score can be validly calculated. I would expect a bifactor solution to fit the data best and to be justified theoretically, since different EDs share some basic variance and the specific factors reflect better some EDs in comparison to others. Here is an example for a similar situation for the Eating Attitude Test-26 (EAT-26): Woropay-Hordziejewicz, N. A., Buźniak, A., Lawendowski, R., & Atroszko, P. A. (2022). Compulsive study behaviors are associated with eating disorders and have independent negative effects on well-being: A structural equation model study among young musicians. Sustainability, 14(14), 8617. Maybe the scale would provide a good fit even for a single factor solution (without bifactor), then it would be even better. Either way, being transparent about measurement in this case will strengthen the contribution. 3) Abdel-Khalek questionnaire (AKQ) - can you provide a bit more data on its reliability and validity since it is single item measure? 4) The Kolmogorov-Smirnov and Shapiro-Wilk tests were used to verify whether they followed the normal distribution. Please note that assumption of normality strictly speaking referrers to normality of the distribution of sample statistic not the sample distribution. Since your sample is very large you don't have to worry about the normality in the case of most analyses. I appreciate that the Authors notice that and provide both parametric and non parametric correlations which show that they are mostly convergent. 5) In the conclusions the Authors write that ED's in the workplace are "very common" - that is a bit overstatement since we're talking about 5% of the population. Relatively common would be a more careful conclusion. 6) Due to the nature of the measurement with EDE-Q the Authors discuss the results in relation to the general category of EDs and only indirectly suggest what could be said about specific ED's such as AN or BED. It should be mentioned in the limitations section because lack of formal diagnosis of particular ED, or specific screening for particular ED instead of general ED risk is a major limitation of this study. 7) Lack of statistical significance of trauma and stress in the last model suggests that depression or happiness may be mediating variables in the association between trauma and stress and EDs risk. Maybe it's worth discussing this possibility. Technically it is possible to test it with this data. However, it would be limited due to the cross-sectional nature of the data. To sum up, I think if the Authors provide these improvements and are transparent about the limitations of this study, the paper is a robust contribution on a topic that is still surprisingly rarely investigated and practically addressed.Author Response
Reviewer #1
The paper assessed the frequency of EDs and studied the relationship between EDs and occupational and individual factors. It is a solid paper both in terms of literature overview which is up to date and refers to most important sources, as well as in terms of methodology. Moreover, it refers to important and understudied area of research related to eating disorders in the workplace, and the potential role of workplace interventions to reduce the scale and negative consequences of these problems. However, few issues should be addressed to strengthen the paper:
Response: We believe that review is a very important part of the scientific process, because it allows us to gather the opinions of researchers who, with different skills and different points of view, can not only improve the presentation of the work done but also suggest useful ideas for the future. Research is sharing knowledge. For this reason, we sincerely thank the reviewer for his/her contribution.
- Authors refer to the DSM framework for diagnosing EDs, however, in most countries worldwide WHO's ICD is the used framework. It would be beneficial to indicate how these frameworks are similar or differ.
Response: We have welcomed this fair note, recalling the two classifications and their convergent validity, without too much burdening the introduction, of which another reviewer complains of excessive length. On the contrary, we believe that an explanation is important because this article is addressed not only to nutritionists, who know the topic well, but also to occupational physicians, who have less knowledge because it is not a professional risk. The text now is as follows:
- EDE-Q (Eating Disorder Examination Questionnaire) was used to measure EDs risk. However, EDs are not a uniform category and different components have slightly different weight and meaning in relation to different EDs, for example weight restraint in AN and BED. It would be recommended to show the factorial structure of EDE-Q (as to my knowledge it has not been done for its short version) to confirm that a general score can be validly calculated. I would expect a bifactor solution to fit the data best and to be justified theoretically, since different EDs share some basic variance and the specific factors reflect better some EDs in comparison to others. Here is an example for a similar situation for the Eating Attitude Test-26 (EAT-26): Woropay-Hordziejewicz, N. A., Buźniak, A., Lawendowski, R., & Atroszko, P. A. (2022). Compulsive study behaviors are associated with eating disorders and have independent negative effects on well-being: A structural equation model study among young musicians. Sustainability, 14(14), 8617. Maybe the scale would provide a good fit even for a single factor solution (without bifactor), then it would be even better. Either way, being transparent about measurement in this case will strengthen the contribution.
Response: The reviewer pointed out a very interesting aspect that had not been taken into consideration by the authors of the short form of the EDE-QS questionnaire, relating to the psychometric characteristics of the instrument. Following his/her advice, we studied the collected data by means of factor analysis, and the result would suggest that it could be multidimensional, with a principal component accounting for 40% of variance and other specific factors. A multifactorial solution could provide better elements than the unifactorial one commonly used. We believe that the reviewer's observation indicates a possible direction for the development of studies on the subject. In this article, which another reviewer has criticized for its excessive length, we have limited our discussion to the idea that a multi-factor solution could be useful for specifically investigating individual disorders, as eating disorders (EDs) are not a uniform category and different components may carry slightly different weight and meaning in relation to various disorders. We included this limitation alongside the others in our study. We consider the reporting of the Polish study on students, which we cited, very useful.
- Abdel-Khalek questionnaire (AKQ) - can you provide a bit more data on its reliability and validity since it is single item measure?
Response: We have gladly added to the article some indications on this questionnaire, which is based on a single question, as most studies in the literature [Lukoševičiūtė J, Argustaitė-Zailskienė G, Šmigelskas K. Measuring Happiness in Adolescent Samples: A Systematic Review. Children (Basel). 2022 Feb 8;9(2):227. doi: 10.3390/children9020227], and as is usual in epidemiological investigations on large populations. According to the original article, it has a 0.86 temporal stability, good concurrent validity (indicated by the highly significant and positive correlations between the single item and the Satisfaction with Life Scale and the Oxford Happiness Inventory), good convergent validity (positive correlation with optimism, hope, self-esteem, positive affect, extraversion, and self-ratings of both physical and mental health), and good divergent validity (negative relationships with anxiety, pessimism, negative affect, and sleeplessness). We have added this explanation, in short, in the methods.
- The Kolmogorov-Smirnov and Shapiro-Wilk tests were used to verify whether they followed the normal distribution. Please note that assumption of normality strictly speaking referrers to normality of the distribution of sample statistic not the sample distribution. Since your sample is very large you don't have to worry about the normality in the case of most analyses. I appreciate that the Authors notice that and provide both parametric and non parametric correlations which show that they are mostly convergent.
Response: We thank the reviewer for this clarification.
- In the conclusions the Authors write that ED's in the workplace are "very common" - that is a bit overstatement since we're talking about 5% of the population. Relatively common would be a more careful conclusion.
Response: The above statement is correct. We have modified it in the direction suggested by the reviewer.
- Due to the nature of the measurement with EDE-Q the Authors discuss the results in relation to the general category of EDs and only indirectly suggest what could be said about specific ED's such as AN or BED. It should be mentioned in the limitations section because lack of formal diagnosis of particular ED, or specific screening for particular ED instead of general ED risk is a major limitation of this study.
Response: We agree. This article reported suspected cases resulting from epidemiological investigation. The workers were warned and advised to proceed with specialist diagnostic tests; the occupational physician generally knows the outcome of these tests only one year later, on the new visit. The method involves a lack of formal diagnosis of particular ED. A promising advancement in this type of study could involve conducting investigations focused on specific types of ED rather than assessing general ED risk. We have added these limitations.
- Lack of statistical significance of trauma and stress in the last model suggests that depression or happiness may be mediating variables in the association between trauma and stress and EDs risk. Maybe it's worth discussing this possibility. Technically it is possible to test it with this data. However, it would be limited due to the cross-sectional nature of the data.
Response: We believe that this idea is also an intriguing contribution for future studies. We have added this text to the discussion: "In the hierarchical multivariate analysis, the introduction of depression and happiness in model IV exceeded the significance of trauma and stress in model III. This statistic could mean that these variables mediate the effect of life trauma or occupational stress. Further studies could clarify this aspect”.
To sum up, I think if the Authors provide these improvements and are transparent about the limitations of this study, the paper is a robust contribution on a topic that is still surprisingly rarely investigated and practically addressed.
R.: Thanks for your work.
Reviewer 2 Report
Comments and Suggestions for Authors
Comments to the Authors of manuscript number nutrients-3735621 entitled “Eating disorders in the workplace.”
- 15–16 "Suspected EDs affected 4.9% of workers, with no notable difference in gender. Cases were significantly associated with trauma and emotional factors (anxiety, depression, unhappiness)...."
The lack of difference between the sexes contradicts the results on lines 286–288, where the authors write:
“Compared to their male colleagues, females manifested higher ED scores (5.23±5.39 vs. 3.96±4.83, p<0.001)...”
- 303–306 “Night workers did not have an EDE-Q score significantly different from that of their colleagues (4.48±4.90 vs. 4.83±5.32, p=0.199).”
“The smokers had a slightly higher EDE-Q score than non-smokers (4.77±5.26 vs. 4.72 vs. 5.19), but the difference was not significant (Student’s t=0.852).”
The values of “4.72 vs. 5.19” are not consistent with the previous sentence. Furthermore, p=0.852 at these values does not seem convincingly “not significant” without reporting the standard deviation – the full statistical context is missing.
- 185 “In this questionnaire, a score of over 15 points can indicate eating disorders.”
The authors do not state whether the cut-off was validated in this shortened version of the EDE-Q (12-item) in the Italian working population. The cited work [76] should include this justification. The validation of the EDE-Q cut-off >15 for the 12-item version should be explained.
- 154–156 “The frequency of EDs was higher among night workers than among their daytime colleagues.”
“The frequency of EDs was higher among smokers than among the non-smoker workers;”
Both hypotheses were proposed even though the literature cited later in the manuscript (e.g. line 424) indicates that no such relationship was found. Therefore, there is no solid initial justification for these hypotheses. The hypotheses regarding night work and smoking should be removed or reformulated.
- Line 323–324 “...health literacy maintained a protective role, even if it did not reach the level of significance.” - far-fetched
Author Response
Comments to the Authors of manuscript number nutrients-3735621 entitled “Eating disorders in the workplace.”
- 15–16 "Suspected EDs affected 4.9% of workers, with no notable difference in gender. Cases were significantly associated with trauma and emotional factors (anxiety, depression, unhappiness)...."
The lack of difference between the sexes contradicts the results on lines 286–288, where the authors write:
“Compared to their male colleagues, females manifested higher ED scores (5.23±5.39 vs. 3.96±4.83, p<0.001)...”
Response. We thank the reviewer because this observation has given us the opportunity to clarify an interesting aspect. The reviewer noted an apparent contradiction, and we thank him/her for this. The article refers to two different things, the mean score of the questionnaire (EDE-QS) and the number of cases exceeding the cutoff (EDs). The mean score in women is higher than in men, but the number of cases is not significantly higher. We have specified better in the text to prevent readers from falling into the same misunderstanding. The interpretation of this result is not difficult. Women have a greater attention to body image and nutrition, and this justifies the higher score compared to men, but the cases of clinical relevance are not significantly different in the two sexes. We have added this comment in the Discussion. The text now is as follows: “The prevalence of EDs cases did not differ across work sectors (Pearson chi square = 0.374) but was slightly higher in females than in males (62 suspected cases, 5.3% CI95% 4.1; 6.7 in female workers vs.31 cases, 4.2%, CI95% 2.9;5.9 in male), although the difference between genders was not significant (Pearson’s chi square =1.234, two-sided p=0.267).” Below, speaking about correlations between variables: “Compared to their male colleagues, females manifested higher EDE-QS scores (5.23±5.39 vs. 3.96±4.83, Student’s t p<0.001, Mann-Whitney U p<0.001), higher anxiety (3.11±2.72 vs. 2.08±2.40, p<0.001), a higher risk of depression (1.98±2.26 vs. 1.33±1.96, p<0.001), and a more elevated disordered sleep score (6.13±3.48 vs. 5.49±3.07, p<0.001).” Finally, in the Discussion: “In our investigation, suspected ED prevalence was not found to be significantly associated with gender. In fact, women reported a higher mean EDE-QS score than men but did not have a significantly higher percentage of suspected ED cases. Most likely, women have a greater attention to body image and nutrition, and this justifies the higher EDE-QS score compared to men, but the cases of clinical relevance were not significantly different in the two sexes. This finding agrees with a study conducted in a German population indicating that EDs are slightly more frequent in females than in males during adolescence, but no difference is observed in adults”.
- 303–306 “Night workers did not have an EDE-Q score significantly different from that of their colleagues (4.48±4.90 vs. 4.83±5.32, p=0.199).”
“The smokers had a slightly higher EDE-Q score than non-smokers (4.77±5.26 vs. 4.72 vs. 5.19), but the difference was not significant (Student’s t=0.852).”
The values of “4.72 vs. 5.19” are not consistent with the previous sentence. Furthermore, p=0.852 at these values does not seem convincingly “not significant” without reporting the standard deviation – the full statistical context is missing.
Response. Thanks to the reviewer who caught a typo, there is a vs. that actually must be ±. The value 5.19 is the standard deviation. We promptly corrected the error and explained the results better, reporting the results of the statistical tests in greater detail. Considering the non-normal distribution of the variables, even if this aspect is negligible due to the large number of observations, we performed comparisons with both parametric and non-parametric tests to confirm the absence of variations in the results. The manuscript now is as follows: “Researchers have hypothesized that night work could be an occupational factor responsible for EDs. Our findings did not confirm this association. In our observations, however, there was an association between EDs and poor sleep quality. This apparent contradiction has some explanations. First, it is necessary to remember the difference that exists between "night worker" and "worker who performs part of the work during the night hours." According to Italian law and scientific evidence, a night worker is someone who performs at least three hours of his/her daily working time between midnight and five in the morning for a minimum of eighty working days per year. In the sample we examined, there were no workers included in this definition because the number of night shifts was lower. The alteration of biorhythms and the interactions with chronotypes that are reported in night workers are unlikely to occur in people who perform a limited number of night shifts. Furthermore, the company doctor's decisions in previous years had excluded workers with health problems from night work. Health surveillance may have determined a healthy worker effect, since workers who work night shifts were in better health than others. This set of factors could explain why EDs were not associated with night work but with the quality of sleep.”.
- 185 “In this questionnaire, a score of over 15 points can indicate eating disorders.”
The authors do not state whether the cut-off was validated in this shortened version of the EDE-Q (12-item) in the Italian working population. The cited work [76] should include this justification. The validation of the EDE-Q cut-off >15 for the 12-item version should be explained.
Response. The reviewer correctly noted that The Eating Disorder Examination Questionnaire (EDE-Q) was originally developed to be applied in the context of clinical practice and treatment. The shortened 12-question version (EDE-QS) that we used was developed specifically for epidemiological screening. This version has a cutoff of 15 points, as suggested by the article we cited [76] “Prnjak, K.; Mitchison, D.; Griffiths, S.; Mond, J.; Gideon, N.; Serpell, L.; Hay, P. Further development of the 12-item EDE-QS: identifying a cut-off for screening purposes. BMC Psychiatry. 2020;20(1):146. doi: 10.1186/s12888-020-02565-5”.
- 154–156 “The frequency of EDs was higher among night workers than among their daytime colleagues.”
“The frequency of EDs was higher among smokers than among the non-smoker workers;”
Both hypotheses were proposed even though the literature cited later in the manuscript (e.g. line 424) indicates that no such relationship was found. Therefore, there is no solid initial justification for these hypotheses. The hypotheses regarding night work and smoking should be removed or reformulated.
Response: The association between night work and EDs is well known. We had reported several articles indicating this association [lines 107-127] and based on these studies we formulated hypothesis 3. The results did not confirm this hypothesis. We already discussed this result in the first version of the paper. In this new version of the article, we have explained in more detail the possible reasons why there is no association between EDs and night work, but there is an association between EDs and poor sleep quality.
Regarding smoking, in the introduction we had given an account of the possible association, see line 79 and following: "Individuals with BED and AN are significantly more likely to be life-time smokers than healthy controls [36]." On the basis of this report, we had formulated hypothesis 4, which however was not confirmed by observations.
- Line 323–324 “...health literacy maintained a protective role, even if it did not reach the level of significance.” - far-fetched
Response: Why far-fetched? This is exactly the result. Health literacy is positively associated with health and in this case inversely associated with the presence of EDs. However, in a multivariate model there are factors more strongly associated with EDs than literacy.
Reviewer 3 Report
Comments and Suggestions for Authors
- Overall Assessment
This manuscript addresses a significant yet underexplored public health issue: the prevalence, associated factors, and metabolic consequences of eating disorder (ED) symptoms among working adults. The authors conducted a survey of 1,912 Italian workers (a response rate of 91.7%) using well-validated questionnaires and basic clinical assessments. They identified a 4.9% prevalence of suspected EDs, predominantly binge-eating spectrum disorders. Using hierarchical logistic regression, they explored occupational, psychological, and metabolic correlates. Given its focus on nutrition-related outcomes in a workplace setting, the paper aligns with the scope of Nutrients and has potential to inform occupational health interventions. However, several conceptual, methodological, and presentational issues need to be resolved before it can be considered for publication.
- Major Comments
# |
Issue |
Recommendation |
2.1 |
Causal language in cross-sectional design– Throughout the manuscript (e.g., conclusion line 538: “Work-related stress plays a role…”), the authors imply causality, despite the observational nature of the study. |
Revise all instances of causal or directional language such as “impact,” “determinants,” or “predictors” to clarify that only associations—not causal relationships—can be inferred from this data. |
2.2 |
Sampling frame and generalizability concerns– Participants were recruited during routine occupational health checks, but the composition of the sample in terms of industry sectors, employment types, and geographic regions is not clearly described. |
Include a participant flowchart and detailed description of recruitment procedures, including company selection criteria, inclusion/exclusion rules, and comparisons between participants and non-participants. Address possible healthy-worker bias or other forms of selection bias. |
2.3 |
Definition of ED cases– The use of an EDE-Q short-form score >15 as a cut-off for suspected EDs lacks validation support, as this threshold was not established in the original validation studies. |
Cite relevant validation literature for the 12-item EDE-Q or conduct ROC analysis within your dataset to justify the threshold. Additionally, perform sensitivity analyses using the continuous EDE-Q score. |
2.4 |
Hierarchical regression approach– The four-step regression model includes variables that are conceptually overlapping (e.g., ERI stress and sleep quality), potentially introducing multicollinearity. No diagnostic tests for collinearity are provided. |
Present correlation matrices and variance inflation factor (VIF) or tolerance statistics. Consider alternative modeling approaches such as parallel mediation models or structural equation modeling to better distinguish direct and indirect effects. |
2.5 |
Null result for shift work hypothesis– The absence of an association between night work and ED risk (contrary to H3) is not adequately discussed. |
Elaborate on the statistical power for subgroup analyses and consider whether misclassification (e.g., rotating vs. fixed night shifts) or lack of meal-timing data may have influenced results. |
2.6 |
Metabolic outcomes interpretation– Suspected ED cases showed higher BMI, dyslipidemia, and metabolic syndrome, but the direction of these associations is ambiguous (ED leading to metabolic issues vs. shared risk factors). |
Expand the discussion to include plausible biological and behavioral mechanisms linking disordered eating and metabolic disturbances. Reference longitudinal evidence where available, particularly regarding binge-eating disorder and metabolic syndrome. |
2.7 |
Lack of statistical detail– Percentages are often reported without denominators, and p-values are presented without corresponding effect sizes (ORs and CIs). |
For greater transparency, include absolute numbers and 95% confidence intervals for all prevalence estimates and regression coefficients in both the text and tables. |
2.8 |
Manuscript length and structure– At over 8,000 words, the manuscript is excessively long, with repetitive background sections (e.g., lines 60–120 vs. 360–430). |
Condense the Introduction and Discussion sections by focusing on literature directly relevant to the study’s aims and findings. Aim for a final word count of ≤6,500, including references. |
- Minor Comments
- Line numbering – Remove line numbers in the final submission, as they disrupt readability in the PDF.
- Abstract – Replace vague expressions like “cases were significantly associated…” with specific quantitative results (e.g., prevalence rates, key ORs).
- Acronyms – Define acronyms such as BED (Binge Eating Disorder) and BN (Bulimia Nervosa) upon first mention in both the Abstract and main text.
- Tables – Ensure Tables 1–3 include complete titles, footnotes explaining scoring systems, cut-offs, and sample sizes.
- References – Update citations to include recent systematic reviews on workplace eating disorders (no sources after 2023 are currently cited).
- Language editing – A professional English language edit is advised to correct minor grammatical errors and improve overall clarity and conciseness.
- Ethical considerations – Clarify whether participation was voluntary or mandatory as part of occupational health surveillance, and describe how confidentiality was maintained.
- Data availability – Provide a functional DOI or accession number for the Zenodo repository; simply stating “data are deposited on Zenodo” is insufficient.
- Final Recommendation
Major Revision
The study presents a valuable contribution to the understanding of eating disorder symptoms in the workplace context. However, the authors must address the methodological limitations outlined above, revise any causal implications, and substantially shorten the manuscript. I would be happy to re-review a thoroughly revised version.
Author Response
Reviewer #3
- Overall Assessment
This manuscript addresses a significant yet underexplored public health issue: the prevalence, associated factors, and metabolic consequences of eating disorder (ED) symptoms among working adults. The authors conducted a survey of 1,912 Italian workers (a response rate of 91.7%) using well-validated questionnaires and basic clinical assessments. They identified a 4.9% prevalence of suspected EDs, predominantly binge-eating spectrum disorders. Using hierarchical logistic regression, they explored occupational, psychological, and metabolic correlates. Given its focus on nutrition-related outcomes in a workplace setting, the paper aligns with the scope of Nutrients and has potential to inform occupational health interventions. However, several conceptual, methodological, and presentational issues need to be resolved before it can be considered for publication.
Response: We thank the reviewer for the numerous and precise observations, which have contributed to significantly improving the text and clarifying the meaning of the results.
- Major Comments
# |
Issue |
Recommendation |
|
2.1 |
Causal language in cross-sectional design– Throughout the manuscript (e.g., conclusion line 538: “Work-related stress plays a role…”), the authors imply causality, despite the observational nature of the study. |
Revise all instances of causal or directional language such as “impact,” “determinants,” or “predictors” to clarify that only associations—not causal relationships—can be inferred from this data. |
|
|
Response: The reviewer correctly recalled that the study, of cross-sectional nature, cannot infer causality. We underlined this concept by recalling in the Discussion that the nature of the epidemiological design "prevented us from recognizing the direction of the associations observed. To interpret the data, we had to rely on the literature."[lines 525-528]. We were very careful not to express causal evaluations, and we revised the manuscript accordingly. The term "impact" is used several times in the manuscript, always with reference to the results of the literature. The terms "predictor" and "determinant" have been used in their statistical sense when inserted into regression equations. We did not find in the conclusion (nor in other parts of the manuscript) the sentence indicated by the reviewer. |
||
2.2 |
Sampling frame and generalizability concerns– Participants were recruited during routine occupational health checks, but the composition of the sample in terms of industry sectors, employment types, and geographic regions is not clearly described. |
Include a participant flowchart and detailed description of recruitment procedures, including company selection criteria, inclusion/exclusion rules, and comparisons between participants and non-participants. Address possible healthy-worker bias or other forms of selection bias. |
|
Response: As indicated in paragraph 2.1, the sample is composed of " All workers exposed to occupational risks who were called for a periodic medical examination designed to prevent risks”. The workers worked in Lazio, in companies monitored by the Università Cattolica del Sacro Cuore in the year 2022. We added this notation [Line 174-175]. We also added in the manuscript this explanation: “The project design was a cross-sectional census. We invited all workers to participate, without any exclusion criteria or obligation to do so. Participation was not encouraged, but it has always been very high in this type of initiative, since neither workers nor companies sustain any expense, because the proposed activities fall within the duties of the occupational physician. Participants signed an informed consent form and authorized the management of their data for scientific purposes as well. The results were transmitted to each employer, to the company prevention services, and to the workers' representatives in anonymous form, together with the occupational physician's advice on the prevention measures to be adopted in the workplace.”. The categories to which the workers belonged are reported in the article. The information provided is kept confidential by the doctor. There was no incentive to participate, but the participation rate was very high. Consequently, a selection bias can be excluded. The healthy worker effect is notoriously present in all samples drawn from workplaces. We have raised the possibility of a healthy worker effect when talking about workers who work night shifts. |
|||
2.3 |
Definition of ED cases– The use of an EDE-Q short-form score >15 as a cut-off for suspected EDs lacks validation support, as this threshold was not established in the original validation studies. |
Cite relevant validation literature for the 12-item EDE-Q or conduct ROC analysis within your dataset to justify the threshold. Additionally, perform sensitivity analyses using the continuous EDE-Q score. |
|
Response: The reviewer correctly noted that The Eating Disorder Examination Questionnaire (EDE-Q) was originally developed to be applied in the context of clinical practice and treatment. The shortened 12-question version (EDE-QS) that we used was developed specifically for epidemiological screening. This version has a cutoff of 15 points, as suggested by the article we cited [76] “Prnjak, K.; Mitchison, D.; Griffiths, S.; Mond, J.; Gideon, N.; Serpell, L.; Hay, P. Further development of the 12-item EDE-QS: identifying a cut-off for screening purposes. BMC Psychiatry. 2020;20(1):146. doi: 10.1186/s12888-020-02565-5”. Readers can find in this article the statistical tests requested by the reviewer. |
|||
2.4 |
Hierarchical regression approach– The four-step regression model includes variables that are conceptually overlapping (e.g., ERI stress and sleep quality), potentially introducing multicollinearity. No diagnostic tests for collinearity are provided. |
Present correlation matrices and variance inflation factor (VIF) or tolerance statistics. Consider alternative modeling approaches such as parallel mediation models or structural equation modeling to better distinguish direct and indirect effects. |
|
Response: Before submitting the article, we verified that the proposed calculations were correct. In the multiple regression analyses we conducted collinearity diagnostics tests, calculating the variance inflation factor (VIF) for each of the multiple linear regression models that we summarized in Table 3. All the VIF values were found to be well below the conventionally adopted limits for the risk of collinearity. We can therefore exclude that there was any problem of multicollinearity in the interpretation of the results. We have added as a supplement in Tables A1 and A2 the collinearity statistics, including tolerance, VIF, eigenvalue and condition index. We have reported in the “Statistics” section of the article the explanation of the statistical method used, and in section 3 a brief synthesis of the results. In statistics there are numerous methods, each of which specifically responds to a specific need. Hierarchical regression models allowed us to follow the overlapping of phenomena, according to a logical order. The search for moderators or mediators was not among the objectives of this work. We believe that the reviewer's suggestion is very useful for future research development. It would be very interesting to know if any of the factors associated with EDs interact with others; this would be of great significance in a longitudinal study, which we intend to develop on the basis of this first cross-sectional data collection. |
|||
2.5 |
Null result for shift work hypothesis– The absence of an association between night work and ED risk (contrary to H3) is not adequately discussed. |
Elaborate on the statistical power for subgroup analyses and consider whether misclassification (e.g., rotating vs. fixed night shifts) or lack of meal-timing data may have influenced results. |
|
Response: We thank the reviewer for giving us the opportunity to delve into an interesting aspect of this study, which was not sufficiently analyzed in the first version of the manuscript. Indeed, as he/she has intuited, there may be an incorrect classification, with the confusion between night worker and employees who also work night shifts. Very often, when we talk about night work, we simply mean work done at night, without considering that, to define this activity according to Italian law, specific requirements must be met. In the reviewed version of the paper, we have written [Line 446 and following]: “Researchers have hypothesized that night work could be an occupational factor responsible for EDs. Our findings did not confirm this association. In our observations, however, there was an association between EDs and poor sleep quality. This apparent contradiction has some explanations. First, it is necessary to remember the difference that exists between "night worker" and "worker who performs part of the work during the night hours." According to Italian law and scientific evidence, a night worker is someone who performs at least three hours of his/her daily working time between midnight and five in the morning for a minimum of eighty working days per year. In the sample we examined, there were no workers included in this definition, because the number of night shifts was lower. The alteration of biorhythms and the interactions with chronotypes that are reported in night workers are unlikely to occur in people who perform a limited number of night shifts. Furthermore, the company doctor's decisions in previous years had excluded workers with health problems from night work. Health surveillance may have determined a healthy worker effect, since workers who work night shifts were in better health than others. This set of factors could explain why EDs were not associated with night work but with the quality of sleep.”. |
|
||
2.6 |
Metabolic outcomes interpretation– Suspected ED cases showed higher BMI, dyslipidemia, and metabolic syndrome, but the direction of these associations is ambiguous (ED leading to metabolic issues vs. shared risk factors). |
Expand the discussion to include plausible biological and behavioral mechanisms linking disordered eating and metabolic disturbances. Reference longitudinal evidence where available, particularly regarding binge-eating disorder and metabolic syndrome. |
|
Response: We thank you for your observation which has given us the opportunity to broaden the discussion on a topic of great interest in public health. We added a longitudinal study to the ones previously mentioned and reported some new references. The paragraph now is as it follows: “In our case studies, an association was observed between EDs and hyperglycemia. This association has also been found in the literature. Eating problems have been reported in patients with type I diabetes [163,164]. US adults with EDs participating in the National Epidemiological Survey on Alcohol and Related Conditions III (NESARC-III) manifested a number of comorbidities. The main ones were depression and alcohol abuse, but hyperglycemia, hypertension and dyslipidemia were also present [165]. Longitudinal studies have shown that BN and BED induce hypertension [166]. Our findings corroborated this association. All these associations between EDs and components of the metabolic syndrome (MetS) that by definition includes at least three out of five disorders (obesity, hypercholesterolemia, hypertriglyceridemia, hyperglycemia, hypertension), explain the association between EDs and MetS we observed and that is present in the literature. Increased inflammation and adipokine dysfunction have been observed in adolescents with BED and obesity [167]. The Brazilian Longitudinal Study of Adult Health (ELSA-Brasil) conducted on more than 15,000 people showed that BED is followed by MetS [168]. A longitudinal study observed that after 5 years of follow-up, people with BED have a more than doubled risk of developing new components of MetS [169]. The studies showed that BED is closely linked to MetS and its components, like high blood pressure, obesity, type 2 diabetes, and unhealthy cholesterol levels, through shared behaviors, genes, and biological factors [170].. |
|||
2.7 |
Lack of statistical detail– Percentages are often reported without denominators, and p-values are presented without corresponding effect sizes (ORs and CIs). |
For greater transparency, include absolute numbers and 95% confidence intervals for all prevalence estimates and regression coefficients in both the text and tables. |
|
Response: We calculated the prevalence and used the exact binomial test with the exact Clopper–Pearson 95% Confidence Interval to measure the degree of uncertainty in prevalence. We have therefore included confidence limits for the proportions in both the text and the tables. The confidence intervals were naturally quite small, given the high number of observations. We reviewed the tables, where there were sometimes typos or missing information, adding number of observations, ORs and CI95%. |
|||
2.8 |
Manuscript length and structure– At over 8,000 words, the manuscript is excessively long, with repetitive background sections (e.g., lines 60–120 vs. 360–430). |
Condense the Introduction and Discussion sections by focusing on literature directly relevant to the study’s aims and findings. Aim for a final word count of ≤6,500, including references. |
|
|
|||
Response: We are sure that the article is longer than other papers on this topic and those we've written on other topics. The greater length is because we have addressed a little-frequent topic, and we have therefore been forced to explain many things that may seem banal for specialists but are little known for occupational physicians. Some parts of the introduction, dedicated to the economic impact of EDs and that seem out of context, are aimed at motivating occupational physicians, who generally do not consider EDs a work-related problem. The considerations in lines 60-120, pleonastic for a nutritionist, are of interest to those who are facing these problems for the first time. We respectfully point out that the request to reduce the article length conflicts with our need to provide an explanation of statistical procedures that we previously omitted for brevity, as well as to expand the discussion on topics such as night work and the metabolic relationships of eating disorders (EDs). We have therefore tried to summarize all the responses as much as possible without losing clarity. Furthermore, it seems to us that reducing the article to 6500 words is not feasible because the references alone are over 6000 words. |
- Minor Comments
- Line numbering – Remove line numbers in the final submission, as they disrupt readability in the PDF.
Response: We used the template provided by the magazine to compose the article. It includes the line numbers.
- Abstract – Replace vague expressions like “cases were significantly associated…” with specific quantitative results (e.g., prevalence rates, key ORs).
Response: we gladly accept this suggestion. We have included in the abstract the prevalence values with confidence intervals and the results of the multivariate analysis with ORs and CI95%.
- Acronyms – Define acronyms such as BED (Binge Eating Disorder) and BN (Bulimia Nervosa) upon first mention in both the Abstract and main text.
Response: All acronyms were defined at the first indication. Also, a list of acronyms is given at the end of the article.
- Tables – Ensure Tables 1–3 include complete titles, footnotes explaining scoring systems, cut-offs, and sample sizes.
Response: We gladly accepted the reviewer's suggestion, making the tables more detailed and understandable. We added number of cases, methods used, notes
- References – Update citations to include recent systematic reviews on workplace eating disorders (no sources after 2023 are currently cited).
Response: In the previous version, the test included 19 articles published in 2024 and 4 published in 2025. The articles published in 2023 that we included in the references were 20. Overall, references from the period 23-25 were 26.9% of all cited studies. We recall that on PubMed the entry “eating disorders” provides over 59 thousand entries, of which about 3500 were published in 2023. The search with the phrase “eating disorders and workplace” provides only 4 entries in 2023, one of which is in Japanese. The only systematic review on the topic of "eating disorders and workplace" that appears on PubMed is the work by Yetsenga et al. that we have cited. In this new version we have added some more very recent articles. If the reviewer believes that there has been a very important study, we would be grateful if he/she would let us know. We will read it carefully and cite it if it adds elements to our study.
- Language editing – A professional English language edit is advised to correct minor grammatical errors and improve overall clarity and conciseness.
Response: As reported in the acknowledgements, the linguistic revision was entrusted to Prof. E.A. Wright, a native English speaker and former professor of scientific English at the Università Cattolica del Sacro Cuore in Rome. We asked the professor to review the article, which will inevitably determine an extension of the re-submission times.
- Ethical considerations – Clarify whether participation was voluntary or mandatory as part of occupational health surveillance, and describe how confidentiality was maintained.
Response: Participation in health surveillance is mandatory according to European regulations adopted in our country for all workers exposed to occupational risks. During the visit scheduled for annual surveillance, workers were invited to participate in a health promotion activity, which was free and not incentivized. Participation was very high in percentage terms. The main reason for non-participation was lack of time to answer the questionnaire. Some respondents were excluded because their answers were incomplete. All those who gave complete answers were included in the study. The data collected are subject to professional secrecy required for medical acts by the Hippocratic Oath and by the confidentiality clause provided for by the ICOH code of ethics for prevention operators and by Italian laws.
- Data availability – Provide a functional DOI or accession number for the Zenodo repository; simply stating “data are deposited on Zenodo” is insufficient.
Response: We regret that the DOI number of the file deposited on Zenodo did not appear on the manuscript submitted the first time. We have added it.
- Final Recommendation
Major Revision
The study presents a valuable contribution to the understanding of eating disorder symptoms in the workplace context. However, the authors must address the methodological limitations outlined above, revise any causal implications, and substantially shorten the manuscript. I would be happy to re-review a thoroughly revised version.